

**Simulating avalanche-triggered lake overspill and downstream impacts at**
**Birendra Lake using RAMMS and HEC-RAS**
Sujan Thapa[1], Ragini Vaidya[1], Mohan Bahadur Chand[1,2*], Rijan Bhakta Kayastha[1,2]
[1]Department of Environmental Science and Engineering, School of Science, Kathmandu
University, Dhulikhel, Nepal
[2]Department of Environmental Science and Engineering, Himalayan Cryosphere, Climate and
Disaster Research Centre (HiCCRDC), School of Science, Kathmandu University, Dhulikhel,
Nepal
* Correspondence: mohan.chand@ku.edu.np





**ABSTRACT**
The study presents the first comprehensive quantitative assessment of avalanche-triggered GLOF
hazards at Birendra Lake using integrated RAMMS-HEC-RAS modelling to evaluate cascading
risks from avalanche release to downstream flood propagation. Three scenarios representing small
($5.1 \times 10^4$ m³), medium ($5.3 \times 10^5$ m³), and large ($1.2 \times 10^6$ m³) avalanche releases from steep
slopes (30°-48.8°) surrounding the lake were simulated. The modelling framework demonstrates
that all scenarios reach Birendra Lake with substantial mass retention (62-86%), generating
maximum velocities of 33.8-72.8 m/s and flow heights of 11.2-36.8 m. The displacement-driven
overspill mechanism displaces 0.01-0.18% of total lake volume ($4.7 \times 10^6$ m³), producing peak
discharge rates of 615.7-3,151.8 m³/s. HEC-RAS flood modelling reveals rapid downstream
propagation, with flood arrival times of 0.15-0.43 hours at Samagaon and 4.6-19.76 hours at Jagat,
accompanied by maximum flood depths of 0.96-12.69 m and velocities of 1.94-15.62 m/s. The
modelling results demonstrate strong qualitative alignment with the April 2024 event, validating
the overspill mechanism. Medium to large avalanche scenarios pose severe threats to downstream
communities, with the large scenario producing catastrophic conditions at Samagaun, where depths
exceed 12 m with velocities above 15 m/s. The findings establish Birendra Lake as an imminent
high-risk system where steep avalanche-prone terrain, lake proximity to unstable glacier zones, and
significant downstream exposure create catastrophic cascading hazards. This research provides
essential quantitative foundations for early warning systems and risk reduction strategies in
avalanche-prone glacial lake environments across High Mountain Asia.
**Keywords:** Glacial Lake Outburst Flood, RAMMS, HEC-RAS, Birendra Lake, Climate Change



# 1. INTRODUCTION

## 1.1. Background

Global climate change is profoundly altering high-mountain environments, most notably through the accelerated retreat of glaciers and the associated formation and expansion of glacial lakes— trends especially pronounced in the Himalayas (Chand & Watanabe, 2018; Clague & Evans, 2000; Maskey et al., 2020). Recent research demonstrates ongoing climate warming and associated glacial lake expansion in the Himalayan region, with documented increases in lake area and corresponding glacier retreat over recent decades (Khadka et al., 2022). These lakes, frequently dammed by unstable moraine or ice barriers (Costa & Schuster, 1988), pose a growing risk of Glacial Lake Outburst Floods (GLOFs) to downstream communities and infrastructure (Worni et al., 2014). Several mechanisms can trigger GLOFs, among which mass movements—such as ice or snow avalanches impacting the lake surface—are particularly important (Emmer & Cochachin, 2013, Schneider et al., 2014). The impulse waves generated by such impacts have the potential to overtop or breach the impounding dam, leading to the sudden and catastrophic release of lake water (Heller et al., 2009).

A stark illustration of this cascading hazard occurred at Birendra Lake on 21 April 2024, when a massive ice-debris avalanche from the Manaslu Glacier triggered significant lake overtopping and downstream flooding (Maharjan et al., 2024). This event demonstrated the vulnerability of the lake system to avalanche impacts from the steep and heavily crevassed glacier snout, generating displacement waves that caused overspill and affected multiple settlements in the Budhi Gandaki valley (Maharjan et al., 2024). The terrain surrounding Birendra Lake exhibits high avalanche susceptibility, with steep slopes (>30°) dominating the upper basin and creating multiple potential release zones (Chaulagain et al., 2025). Understanding the hazard sequence from initial avalanche dynamics to potential lake overspill and downstream flooding in the Budhi Gandaki River valley is crucial for effective risk assessment and mitigation (Worni et al., 2015; Richards & Reddy, 2007).

Numerical modelling is essential in analysing and simulating these complex cascading processes (Worni et al., 2014). This investigation proposes the use of established tools, specifically RAMMS (Rapid Mass Movement Simulation):: Avalanche for simulating ice avalanche dynamics (Christen



et al., 2010; Casteller et al., 2008), alongside HEC-RAS (Hydrologic Engineering Center's River
Analysis System) for modeling subsequent hydrological overspill and downstream flood
propagation (Brunner, 2016; Feldman, 2000). The integration of these models enables a
comprehensive simulation of the entire event chain, from avalanche impact to downstream effects
(Worni et al., 2015), while also acknowledging inherent limitations—such as the adaptation of
snow avalanche models for ice avalanches (Bartelt et al., 2012; Gauer et al., 2008) and challenges
related to data scarcity, particularly limited lake bathymetric data (Huss et al., 2017; Østrem &
Brugman, 1991).
**1.2. Ice Avalanche Impact on Glacial Lakes**
When an ice avalanche impacts the surface of a glacial lake, it transfers momentum to the water
body, generating impulse waves (Zitti et al., 2016). The characteristics of these waves, including
their amplitude and velocity, are influenced by several factors such as the volume and velocity of
the impacting mass, the morphology of the lake basin, the Froude number, and the density
difference between the avalanche material and the lake water (Zitti et al., 2016; Walder et al.,
2003). Empirical models developed for landslide-generated waves are often adapted to simulate
avalanche-induced waves; however, the lower density of snow and ice can lead to overestimations
if these models are applied without modification (Zitti et al., 2016). Chisolm and McKinney (2018)
conducted comprehensive three-dimensional (3D) lake-wave simulations for Lake Palcacocha in
Peru, which provided critical insights into these dynamics. The volume released from the lake is
consistently only a fraction (f) of the avalanche's initial ice volume. Specifically, their large-
avalanche scenario resulted in 60% of the avalanche mass overtopping the dam, while medium and
small scenarios yielded 50% and 30% overtopping, respectively. This established a representative
displacement fraction (f) range of approximately 0.3–0.6 for such events. Furthermore, these
avalanche-induced surges are inherently brief. Chisolm and McKinney (2018) reported that the
initial wave overtopping for their large-avalanche case at Lake Palcacocha lasted approximately
100 seconds, with smaller avalanches producing overtopping durations of only 50-70 seconds. This
finding is consistent with other studies indicating that GLOF wave generation by fast debris falls
typically evolves within seconds to a few minutes. A direct consequence of these brief durations is
that shorter pulse durations (T), typically 10–100 seconds (tens to hundreds of seconds), result in
significantly higher peak flows for a given flood volume.
In the Himalayan context, avalanche-generated GLOFs exhibit comparable rapid behaviour,



reinforcing the applicability of these parameters. The 2016 Gongbatongsha event in the
Poiqu/Bhotekoshi basin, triggered by a debris/rock avalanche into a small Tibetan Lake,
demonstrated a modelled lake-emptying time of only a few minutes. Sattar et al. (2022)
reconstructed this flood, with their best-fitting scenario indicating that the $0.12 \times 10^6$ m³ lake
emptied in approximately 6 minutes, reaching a peak discharge of ~620 m³/s just 30 seconds after
initiation. This suggests an exceptionally impulsive release. GLOF models for Imja Tsho in Nepal
frequently employ impulse-wave inputs calibrated by rapid avalanche collapse, with the time from
avalanche entry to terminal moraine run-up and subsequent outlet discharge estimated at
approximately 3 minutes (Lala et al., 2018). Sattar et al. (2021) modelled avalanche impacts on
Lower Barun Lake, reporting very short overtopping pulse durations of 20-21 seconds,
accompanied by substantial peak discharges of 9,298 m³/s and 8,300 m³/s for two distinct avalanche
scenarios, respectively.
The consistent reporting of short pulse durations (ranging from tens to hundreds of seconds, or a
few minutes) and significant displacement fractions across diverse global (e.g., Palcacocha, Peru)
and Himalayan (e.g., Gongbatongsha, Imja Tsho, Lower Barun) case studies provide a robust
empirical and modelling precedent. This strong evidence base rigorously supports applying these
parameters to Birendra Lake, underscoring that avalanche-triggered GLOFs are fundamentally
impulsive events characterised by rapid water release and high peak flows. Accurately capturing
this characteristic is critical for precise hazard assessment.
**1.3. Integrated Avalanche and Hydraulic Modelling for Cascade Hazards**
Integrating avalanche models like RAMMS with hydraulic models like HEC-RAS has been
successfully applied in various case studies to simulate cascade hazards, particularly avalanche-
triggered floods and landslides into lakes (Copernicus Emergency Management Service, 2025).
For example, RAMMS has been used to simulate ice avalanches that subsequently triggered debris
flows, highlighting the potential for cascading events (Mergili et al., 2022). In another study by
Somos-Valenzuela et al. (2018), RAMMS was coupled with a hydrodynamic model
(BASEMENT) to simulate avalanche-induced waves in a glacial lake. This then informed the
simulation of moraine erosion and downstream flooding. These studies often use the outputs from
the avalanche model, such as the volume and velocity of the mass at the point of impact with the
lake, as input conditions for the hydraulic model to simulate the resulting flow or inundation.



HEC-RAS has been widely used to model the downstream flood propagation resulting from glacial
lake outburst floods triggered by various mechanisms, including avalanches (Klimeš et al., 2014).
These studies often involve reconstructing past GLOF events using field surveys and eyewitness
accounts to calibrate the hydraulic models and assess the flood hazard in downstream areas.
Integrated modelling approaches that link RAMMS simulations of avalanche impact with HEC-
RAS simulations of lake overspill and downstream flooding provide a comprehensive framework
for understanding the hazard chain (Somos-Valenzuela et al., 2018). These integrated models are
crucial for identifying vulnerable areas, assessing the potential impacts on infrastructure and
settlements, and developing effective mitigation measures for avalanche-triggered GLOF hazards
(Copernicus Emergency Management Service, 2025).

**1.4. GLOF and Avalanche Hazards in the Nepal Himalayas**

The Nepal Himalayas is highly susceptible to glacial lake outburst floods (GLOFs) and avalanche
hazards due to numerous glaciers and glacial lakes in a seismically active region undergoing rapid
climate change (Carrivick & Tweed, 2016). Climate change is accelerating glacier retreat, leading
to the formation and expansion of glacial lakes, thus increasing the risk of GLOFs (Byers et al,
2020, Carrivick & Tweed, 2016). Avalanches, including snow and ice avalanches, are significant
triggers for GLOFs in this region, often causing displacement waves in glacial lakes that can
overtop or breach moraine dams.
Birendra Lake, located at the base of the Manaslu Glacier in the Gorkha District of Nepal,
experienced a notable flood event on April 21, 2024, which was triggered by a massive ice
avalanche from the glacier snout (Mehar, 2024). This event caused a displacement wave in the
lake, leading to overspill and flooding downstream, destroying a bridge (Fig. 1) is not a typical
GLOF involving a moraine dam breach, highlights Birendra Lake's vulnerability to avalanche-
triggered flooding (Maharjan et al., 2024).
Studies have indicated that even relatively small ice-snow avalanches can generate surge waves in
Birendra Lake, leading to repeated GLOFs. The Manaslu region, in general, is prone to both glacial
lake hazards and avalanches, necessitating further research to understand the potential for future
cascading events (Mehar, 2024). Continuous monitoring of glacial lakes and glacier dynamics in
the Manaslu region is crucial for effective risk assessment and the development of mitigation
strategies. Here, in this paper we aimed estimating the potential cascading hazard posed by ice



avalanches originating from the Manaslu Glacier impacting Birendra Lake and triggering
downstream flooding, using an integrated modelling approach with RAMMS::Avalanche and
HEC-RAS, while explicitly acknowledging model and data limitations. We simulated the plausible
ice avalanche scenarios using RAMMS::Avalanche with adjusted parameters to determine their
runout characteristics and the volume of ice deposited into Birendra Lake. Similarly, we utilised
HEC-RAS to model the hydrological response of Birendra Lake to the simulated ice avalanche
inputs.

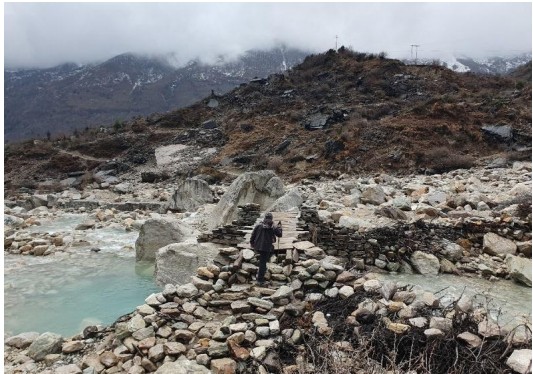


161          **Figure 1** Bridge connecting Samagaun and Samdo (10 April 2025)

**2.  MATERIALS AND METHODS**
**2.1. Study Area**
Birendra Lake is an end-moraine-dammed glacial lake in Chumanubri Rural Municipality, Gorkha
District, Nepal (Fig. 2), occupying roughly 0.24 km² at about 3,632 m asl on the northeast base of
Mount Manaslu (8,163 m). Remote-sensing and field observations confirm that the Manaslu
Glacier has recently detached from direct contact with the lake, leaving a steep, heavily crevassed
snout that is highly susceptible to ice-avalanche release and consequent water displacement
(Maharjan et al., 2024). On 21 April 2024, such an avalanche generated an overspill flood with an
estimated peak discharge of 32 m³/s (Maharjan et al., 2024) that destroyed the downstream
footbridge at Samagaon, demonstrating the cascade hazard from slope instability to riverine
impacts. The downstream areas of focus for this study along the Budhi Gandaki River include
Samagaun (Site 1), Lhi (Site 2), Namrung (Site 3), Ghap (Site 4), Deng (Site 5), and Jagat (Site 6).





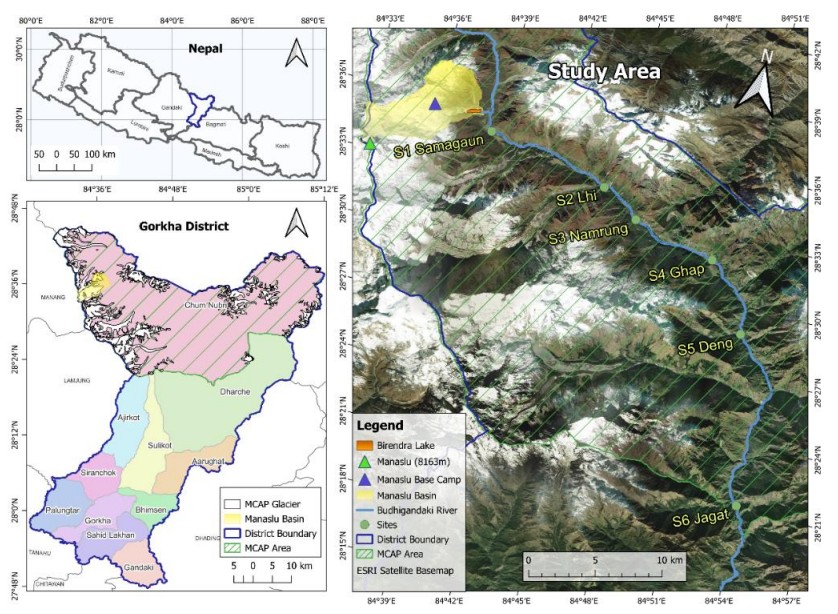


**Figure 2** Study Area Map (© Google Maps 2025)

**2.2. Data collection**

A comprehensive dataset was assembled to support this study's integrated avalanche-flood modelling approach. A 12.5m resolution ALOS PALSAR DEM and a corrected 30 m resolution SRTM DEM were acquired to provide detailed topographic information of the study area, which is essential for accurately simulating surface processes and hydrological modelling. High-resolution optical satellite imagery from Planet Labs (Accessed April 2025) was utilised to estimate the lake's surface area, which is crucial for empirical lake volume calculations. Field observations, including geo-tagged photographs and qualitative insights from local interviews, were gathered during a site visit. The literature review provided the theoretical foundation and parameter calibration guidance for avalanche and flood modelling components. The flowchart of the study is provided in the Fig. 3.

**2.2.1. Identification of Potential Avalanche Release Areas using GIS**

Potential avalanche release zones were identified using the Bühler et al. (2013) multi-criteria methodology implemented in Google Earth Engine. This approach employs a rigorous binary classification system that categorises terrain as susceptible (value = 1) or non-susceptible (value = 0) to avalanche initiation. The methodology applies four simultaneous terrain criteria: slope angle



(28-60°), curvature (≤50), terrain roughness (≤15m standard deviation), and non-forested areas.
Areas must satisfy all four criteria simultaneously to receive a classification of susceptible (1),
while areas failing any single criterion are classified as non-susceptible (0). The Bühler
methodology was originally validated against over 8,000 mapped avalanche release areas across
the Swiss Alps and subsequently tested in the Indian Himalayas (Manali region, Himachal
Pradesh), demonstrating strong transferability to high-mountain Asian environments. Selected
susceptible zones were then filtered by minimum area (>780 m², approximately 5 pixels) to
eliminate small, isolated areas unlikely to generate significant avalanches capable of reaching
Birendra Lake, while ensuring adequate spatial extent for reliable RAMMS numerical
simulation and proximity to Birendra Lake to identify three representative release scenarios for
RAMMS simulation.

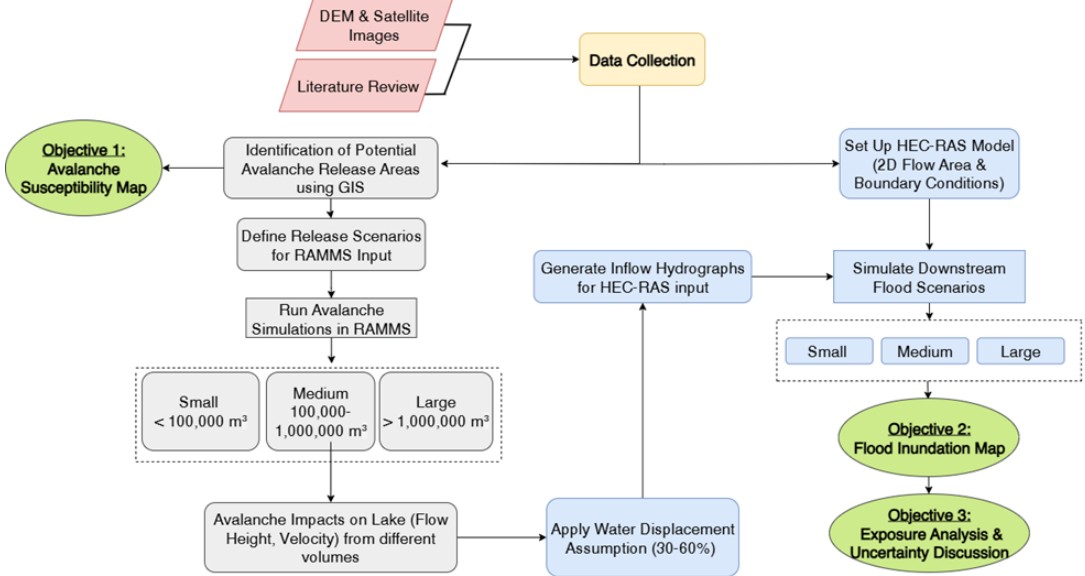


**Figure 3** Flow Chart of Methodology
**2.2.2.  Define Release Scenarios for RAMMS Input**
Ice avalanche scenarios were systematically developed based on identified potential release zones,
and documented ice avalanche volume ranges from high-mountain environments. Recent
comprehensive reviews of avalanche hazards in High Mountain Asia confirm the wide variability
in avalanche magnitudes, with ice detachments from hanging glaciers and seracs capable of
producing high-impact events (Acharya et al., 2023). Three distinct scenarios were defined with




varying release volumes: Small (≤100,000 m³), Medium (100,000-1,000,000 m³), and Large
(>1,000,000 m³). These volume ranges are consistent with documented ice avalanche magnitudes,
where most break-off volumes in edge situations are well below 1 million m³, while ramp situations
can produce volumes exceeding 1 million m³ (Alean, 1985). The classification accounts for the
wide variability in ice avalanche volumes observed in recent events, ranging from $10^3$-$10^5$ m³ for
smaller events to $10^5$-$10^6$ m³ for larger catastrophic events, as documented in recent assessments of
cryospheric hazards in high mountain areas (Hock et al., 2019).
A consistent initial release depth of 5.0 meters was applied for all three scenarios, comparable to
the 4.7m release depth used by Mandal et al. (2025) in the Lower Barun region and consistent with
typical ice failure depths observed in similar Himalayan contexts. This standardised depth ensures
that the substantial increase in volume from the Small to the Large scenario is primarily driven by
expanding release area rather than varying initial thickness of ice failure, providing a systematic
approach to scenario scaling that reflects natural avalanche formation processes.

### 2.2.3. Run Avalanche Simulations in RAMMS

A high-resolution digital elevation model (DEM) was imported into RAMMS::Avalanche to
provide the topographic foundation for simulating avalanche flow paths. Release zones were
systematically delineated for each scenario based on terrain steepness, glacier stability, and
morphological characteristics. The material density was set to 1000 kg/m³, representing
consolidated ice conditions typical of glacial avalanches (Christen et al., 2010; Sattar et al., 2021).
The Voellmy-Salm friction model was used to simulate avalanche dynamics, with a Coulomb
friction coefficient (μ) of 0.12 and a turbulent friction coefficient (ξ) of 1000 m/s², consistent with
parameter values commonly applied in ice avalanche modelling for Himalayan settings. These
parameters were validated by Sattar et al. (2021) for the modelling of lake outburst and downstream
hazard assessment at Lower Barun Glacial Lake and further applied by Mandal et al. (2025) in their
Lower Barun region avalanche studies, supporting their applicability across similar high-mountain
Himalayan environments.

### 2.2.4  Estimating Glacial Lake Volume

In many high-mountain regions worldwide, including the Himalayas, the lack of detailed
bathymetric data for glacial lakes presents a significant challenge for hazard assessment and
hydrological analysis (Huggel et al., 2002). Here, we also used equation proposed by Huggel et al.



(2002), which also adapted in in studies of glacial lakes in other mountainous regions, including
the Himalayas, to estimate lake volumes where direct measurements are unavailable (Yao et al.,
2012; Wang et al., 2018).
$$V = 0.104 \times A^{1.42}$$
Where $V$ represents lake volume (m³) and $A$ denotes surface area (m²).

### 2.2.5  Set Up HEC-RAS Model (2D Flow Area & Boundary Conditions)

Geometric data were prepared using DEM-extracted terrain data in HEC-RAS, representing
Birendra Lake as a storage area and the downstream Budhigandaki River reach. The stage-storage
relationship for the lake was defined based on DEM-derived area and estimated average depth
using empirical area-volume relationships. Manning's roughness coefficient (n) was set to 0.06 for
the main channel, representing typical conditions of Himalayan Mountain streams characterised by
rocky beds, irregular banks, and moderate vegetation. This value falls within the established range
of 0.030-0.070 for natural mountain channels (Chow, 1959). Given the limited field data available
for precise roughness calibration, it provides a conservative estimate appropriate for flood hazard
assessment.

### 2.2.6  Generate Flood Hydrographs from RAMMS Output

A point at the lake boundary with the highest flow height from RAMMS output was selected to
create a time series of avalanche impact. This helped estimate the arrival time of the avalanche, the
duration of lake disturbance, and the timing of peak flows. These values were used to develop a
basic flood hydrograph without simulating detailed wave behaviour. Previous studies show that
such overtopping events are very brief. For example, Chisolm and McKinney (2018) reported
overtopping at Lake Palcacocha lasting about 100 seconds for a large avalanche, while smaller
cases lasted 50 to 70 seconds. In the Himalayas, similar short pulse durations have been recorded.
Sattar et al. (2022) found a 6-to-10-minute lake-emptying time at Gongbatongsha. Lower Barun
Lake had pulse durations of 20 to 21 seconds with very high discharges (Sattar et al., 2021). These
studies also suggest that only 30 to 60 per cent of the avalanche volume contributes to the actual
flood discharge (Chisolm & McKinney, 2018). Although this method does not model impulse
waves directly, it provides a simple and practical estimate of the lake response.




### 2.2.7 Simulate Downstream Flood Scenarios


HEC-RAS unsteady flow simulations were executed for each scenario (Small, Medium, Large) to
simulate downstream flood propagation resulting from lake overspill. The modelling approach
used converted RAMMS discharge hydrographs as inflow boundary conditions, with initial lake
levels adjusted based on estimated water displacement volumes. The methodology assumes that
avalanche-deposited ice displaces a volume of water of a different displacement fraction, leading
to immediate overspill once the adjusted lake level exceeds the dam capacity. While not explicitly
modelling impulse wave generation or complex avalanche-water interaction dynamics, this
simplified approach provides conservative estimates of downstream flood impacts suitable for
hazard assessment purposes (Westoby et al., 2014; Worni et al., 2014). The coupling methodology
follows established mass flow to flood conversion practices in similar hazard assessment studies
(Mergili et al., 2020).

### 2.2.8 Exposure Analysis and Uncertainty Discussion


Results across scenarios were analysed to understand sensitivity to avalanche volume and assess
the cascade effect from ice avalanche to downstream flooding. Model outputs were compared with
the peak-discharge estimates and inundation limits reported in Poudel (2025), providing a
qualitative check that the simulated floods lie within the published range for Birendra Lake. The
uncertainty discussion provides transparency regarding model limitations and guides the
interpretation of results for further application.

## 3    RESULTS AND DISCUSSIONS


### 3.2    Glacial Lake Volume Calculation


Lake area determination from high-resolution satellite imagery yielded a delineated surface area of
0.246 km² (246,000 m²) for Birendra Lake (Table 1). Similarly, application of established empirical
relationships developed by Huggel et al. (2002) produced a total lake volume of $4.7 \times 10^6$ m³ with
an average depth of 19.11 m. These parameters represent critical baseline data for subsequent
avalanche displacement modelling and risk assessment.





**Table 1** Physical characteristics of Birendra Lake derived from Planet Labs satellite imagery
analysis and empirical volume-area relationships

| Parameter | Value | Units |
|-----------|-------|-------|
| Area | 246000 | m² |
| Volume | $4.7 \times 10^6$ | m³ |
| Depth | 19 | m |


The calculated volume provides essential baseline data for avalanche displacement modelling
scenarios, which are presented in subsequent sections. The substantial depth-to-area ratio (19 m
average depth across 0.246 km²) indicates typical glacial lake morphology with significant water
storage capacity characteristic of moraine-dammed systems. This $4.7 \times 10^6$ m³ volume serves as
the reference against which all displacement scenarios are evaluated, representing the total
available water mass for potential overspill events.

The volume estimate's reliability is supported by consistency with regional glacial lake studies
utilising similar empirical approaches, though the methodology inherently carries uncertainties
associated with bathymetric assumptions. The 3 m spatial resolution of the Planet Labs imagery
provides appropriate precision for lake area delineation at this scale. Future direct bathymetric
validation would enhance precision for refined flood modelling applications, particularly for
catastrophic scenario planning, where volume accuracy directly influences downstream hazard
assessment and risk management strategies.

**3.3  Avalanche Susceptibility Mapping**
Avalanche susceptibility mapping for the Birendra Lake catchment area was conducted using the
Bühler et al. (2013) multi-criteria methodology, which applies four simultaneous terrain parameters
to produce binary classification (susceptible/not susceptible): slope angle (28-60°), curvature
(≤50), terrain roughness (≤15m standard deviation), and non-forested conditions. This conservative
approach ensures that only areas meeting all necessary physical conditions for avalanche formation
are classified as susceptible, providing high confidence in release zone identification while
eliminating false positives common in single-parameter assessments.
The susceptibility map (Fig 4) reveals strategically concentrated susceptible zones (red areas)
around steep glacier margins and unstable ice formations adjacent to Birendra Lake, with
the Manaslu climbing route traversing multiple identified susceptible areas. Comparison with



previous slope-based analysis by Chaulagain et al. (2025) demonstrates significant methodological
advantages: while the slope-only approach classified extensive areas using three risk tiers (high:
30-45°, moderate: 10-30° & 45-60°, low: <10° & >60°), the Bühler multi-criteria method produces
more spatially discrete and physically justified susceptible zones.

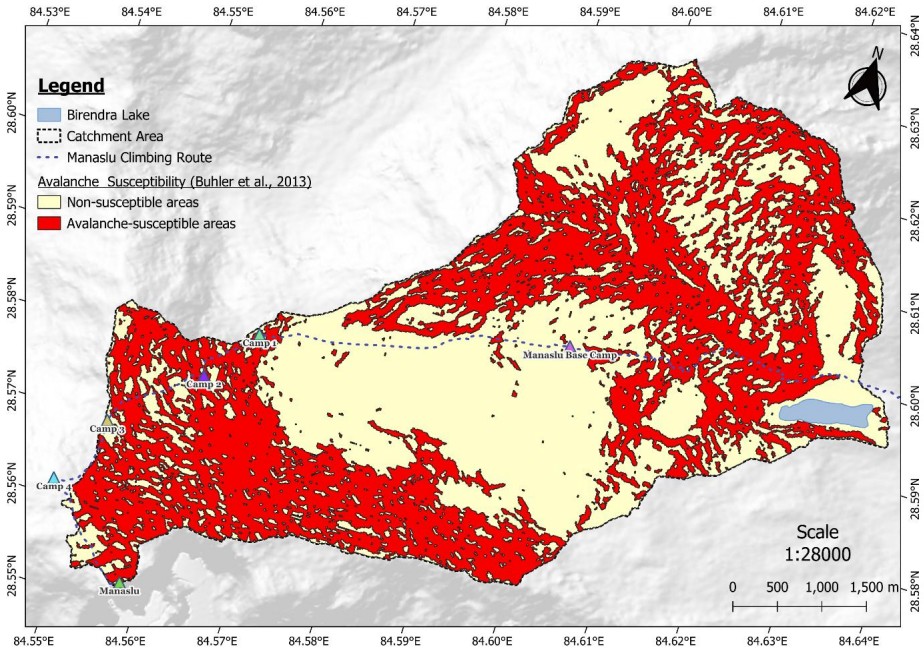


**Figure 4** Avalanche susceptibility map of Birendra Lake catchment area based on Bühler et al.
(2013) multi-criteria analysis, showing susceptible zones (red) and climbing route infrastructure.

The binary classification eliminates ambiguous "moderate risk" categories and focuses on terrain
where all necessary physical conditions for avalanche formation exist simultaneously. This
approach aligns with established practices in glacial hazard assessment, where terrain-based
susceptibility mapping provides the foundation for process-based modelling applications. The
methodology's robustness is particularly relevant for high-altitude environments where complex
topographic interactions govern snow and ice stability.

This analysis systematically selected three representative release scenarios from identified
susceptible polygons (>780 m² minimum area threshold), ensuring that subsequent RAMMS
avalanche simulations originate from terrain with scientifically validated avalanche formation



potential rather than broad slope-based assumptions. The 780 m² minimum area threshold reflects
DEM resolution constraints (12.5 m × 12.5 m pixels) and established best practices for meaningful
avalanche release zone delineation in numerical modelling applications. The climbing route
infrastructure and camp locations were adapted from Adventure Consultants' website (accessed
June 2025), providing critical context for understanding exposure patterns along established
mountaineering routes. This integration of hazard mapping with recreational infrastructure
highlights the practical applications of susceptibility analysis for risk assessment and route
planning in high-mountain environments.

**3.4   Release Zone Selection and Scenario Development**
A preliminary avalanche flow analysis was conducted across the entire Birendra Lake catchment
using a standardised 1-meter release depth, with computational constraints limiting the basin-scale
modelling resolution (Fig 5). The catchment-wide simulation revealed that avalanches originating
from higher elevation zones lost momentum and deposited across extensive non-susceptible terrain
in the central areas of the catchment (complete sequence in Appendix A), indicating minimal direct
lake impact potential from distant release areas. This initial screening process demonstrated that
topographic barriers and extended runout distances significantly attenuate avalanche energy before
reaching the lake vicinity.

Based on these findings, subsequent analysis was refined to focus exclusively on release zones
where avalanche flows demonstrated a clear trajectory toward Birendra Lake, ensuring realistic
avalanche-lake interaction scenarios for downstream flood modelling. This targeted approach
aligns with established best practices in glacial hazard assessment, where process-based modelling
efficiency is optimised through strategic release zone selection rather than exhaustive basin
coverage. The release area parameters for Birendra Lake show notable similarities and differences
compared to other extensively studied Himalayan glacier lake systems. At Lower Barun Lake, the
primary avalanche susceptibility was identified on south-facing slopes where ice-snow masses
hang precariously on slopes between 45-60° (Sattar et al., 2021), comparable to Birendra's large
scenario mean slope of 48.8°. Field observations at Lower Barun documented avalanche volumes
of approximately $1.12 \times 10^5$ m³ of ice-snow, with modelled events reaching $9.2 \times 10^5$ m³ (Sattar et
al., 2021), which falls within the range of Birendra's medium to large scenarios.





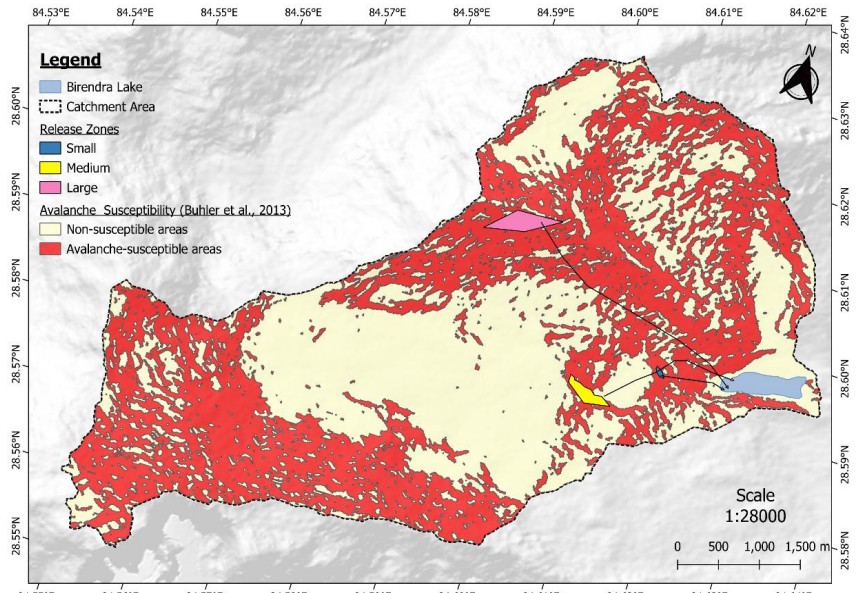

**Figure 5** Potential release zones identification for RAMMS input showing three scenario
classifications: small (blue), medium (yellow), and large (pink) based on Bühler et al. (2013)
susceptibility mapping and preliminary flow trajectory analysis

**Table 2** RAMMS release area properties for three avalanche scenarios derived from
susceptibility mapping and trajectory analysis

| Parameter | Value | | |
|---|---|---|---|
| | **Small** | **Medium** | **Large** |
| Mean slope angle (°) | 43.3 | 44.8 | 48.8 |
| Mean altitude (m) | 4,130 | 4,510 | 5,770 |
| Projected area (m²) | 7,300 | 73,300 | 145,000 |
| Initial Release volume (m³) | 51,200 | 534,000 | 1,165,000 |

Note: Values reflect appropriate precision based on 12.5 m×12.5 m DEM resolution (±156 m²
pixel uncertainty) and established uncertainty propagation principles in glacial hazard
modeling.


At Palcacocha Lake in Peru, avalanche scenarios modelled volumes ranging from $0.15 \times 10^6$ m³
(small) to $1.8 \times 10^6$ m³ (large) (Chisolm & McKinney, 2018), with the large scenario volume being
significantly higher than Birendra's largest modelled event. The Palcacocha study identified release





zones with slopes between 45-60° and applied similar density assumptions of 1000 kg/m³ for ice-
dominated avalanches (Schneider et al., 2014), consistent with the parameters used in the Birendra
analysis. Three distinct ice avalanche scenarios (small, medium, and large) were defined based on
release zones identified through the Bühler et al. (2013) multi-criteria susceptibility analysis to
investigate a range of potential events. The parameters for these scenarios, detailed in Table 2, were
selected to represent systematic progression in event magnitude with physically meaningful scaling
relationships based on observed patterns from comparable Himalayan systems. The Small scenario
simulates a localised release of 51,200 m³ from a 7,300 m² area at a mean altitude of 4,130 m,
representing typical slope instability events common in glaciated high-mountain environments.

The Medium scenario scales significantly, with a release volume of 534,000 m³, representing a
nearly tenfold increase in avalanche magnitude. This scenario originates from a higher elevation
(4,510 m) and steeper terrain (44.8°), reflecting upper glacier zones' enhanced gravitational
potential and slope instability. The Large scenario models a catastrophic event from the highest
and steepest parts of the catchment area, with a release volume of 1,165,000 m³ from a 145,000 m²
area at a mean elevation of 5,770 m and slope angle of 48.8°. For all three scenarios, a consistent
initial release depth of 5.0 meters was applied, comparable to the 4.7-4.9 m depths used by Mandal
et al. (2025) and Sattar et al. (2021) in the Lower Barun region and typical ice failure depths in
similar Himalayan contexts. This standardisation ensures that volume increases from Small to
Large scenarios are driven by expanding release area rather than variable failure thickness,
providing physically consistent scaling relationships. The reported precision acknowledges
inherent DEM-derived uncertainties while maintaining sufficient accuracy for hazard assessment
and downstream flood modelling.

**3.5   RAMMS Simulation Results**
RAMMS avalanche simulations were conducted for the three defined scenarios to quantify flow
dynamics, impact parameters, and lake deposition characteristics. The numerical modelling reveals
fundamental relationships between release volume, flow behaviour, and downstream impact
potential that are critical for understanding avalanche-lake interaction dynamics at Birendra Lake.
**3.5.4   Flow Dynamics and Impact Parameters**
The avalanche simulations demonstrate clear escalation patterns across the three scenarios, with



dynamic parameters showing systematic increases reflecting larger events' enhanced destructive potential. Maximum flow velocities exhibit substantial scaling, ranging from 33.8 m/s for the small scenario to 72.8 m/s for the large scenario, indicating the potential for extremely high-speed ice flows in major avalanche events. These velocities exceed typical threshold values for catastrophic impact and are consistent with observations from similar Himalayan avalanche events. Maximum flow heights increase dramatically across scenarios, from 11.2 m (Small) to 36.8 m (Large), demonstrating the substantial vertical extent of ice avalanche flows. The maximum impact pressures show the most dramatic scaling relationship, ranging from 1,145 kPa in the small scenario to 5,295 kPa in the large scenario. These impact forces can cause severe structural damage and generate significant water displacement upon lake impact, highlighting the destructive potential of larger avalanche events.

The computational error between initial and RAMMS-calculated release volumes remained below 5% across all scenarios, demonstrating acceptable model precision for hazard assessment applications. This level of accuracy is consistent with established RAMMS modelling standards and provides confidence in the reliability of the flow dynamics and impact parameters derived from the simulations.

**Table 3** RAMMS simulation summary showing dynamic parameters and lake deposition characteristics for three avalanche scenarios

| Parameters | Value | | |
|---|---|---|---|
| | **Small** | **Medium** | **Large** |
| RAMMS computed release volume ($m^3$): | 53,100 | 528,000 | 1,171,000 |
| Volume reaching lake ($m^3$): | 45,900 | 328,000 | 845,000 |
| Release volume reaching lake (%): | 86.5 | 62.1 | 72.2 |
| Overall max velocity (m/s): | 33.8 | 48.1 | 72.8 |
| Overall max flow height (m): | 11.2 | 28.2 | 36.8 |
| Overall max pressure (kPa): | 1,150 | 2,320 | 5,300 |



### 3.5.5 Lake Deposition Efficiency

The lake deposition volumes reveal that substantial portions of the released ice material reach Birendra Lake, with absolute volumes ranging from 45,900 m³ (small) to 845,000 m³ (large). However, the percentage of release volume deposited shows non-linear behaviour that provides important insights into avalanche transport mechanics. The small scenario achieves the highest deposition efficiency at 86.5%, the medium scenario shows the lowest at 62.1%, and the large scenario reaches 72.2%. This variation suggests that larger avalanches experience greater material loss during transport due to entrainment processes, deposition along the flow path, or lateral spreading effects. Conversely, smaller avalanches may follow more direct paths to the lake with higher material retention efficiency, possibly due to better topographic confinement and reduced opportunity for material dispersal during transport.

### 3.5.6 Spatial Flow Patterns and Lake Impact Potential

The simulation results reveal distinct spatial patterns that reflect the influence of release volume and topographic constraints on flow behaviour. Figures 6, 7, and 8 illustrate the RAMMS avalanche simulation results, showing distinct differences in flow extent, height distribution, and lateral spreading characteristics across the three scenarios. The estimated release volumes in the figures refer to the initial release volumes, and computational error between initial and RAMMS-calculated release volumes remained below 5% across all scenarios, demonstrating acceptable model precision for hazard assessment applications.

The Small scenario (Fig 6), with a release volume of 53,100 m³, follows the most direct path to Birendra Lake. It remains highly channelised by the natural topography, resulting in a narrow flow corridor that efficiently reaches the lake with minimal lateral spreading (see supplementary Appendix B). This behaviour maximises transport efficiency and explains the higher deposition percentage observed for this scenario. The Medium scenario (Fig 7), originating from a 528,000 m³ release, presents a more constrained but still powerful flow. It remains within the main valley system but exhibits significant lateral spreading and material loss along its path before terminating in the lake. This scenario represents a transitional behaviour between fully confined and unconfined flow regimes (see supplementary Appendix C). The Large scenario (Fig 8), with a release volume of 1,171,000 m³, generates the most extensive flow pattern. Its immense volume and high starting elevation cause the flow to impact Birendra Lake and spill over an eastern ridge, creating a broad




path with lateral spreading exceeding 1 km (see supplementary Appendix D). This multi-
directional flow behaviour demonstrates how catastrophic events can exceed natural topographic
confinement, potentially affecting areas beyond the primary drainage basin.

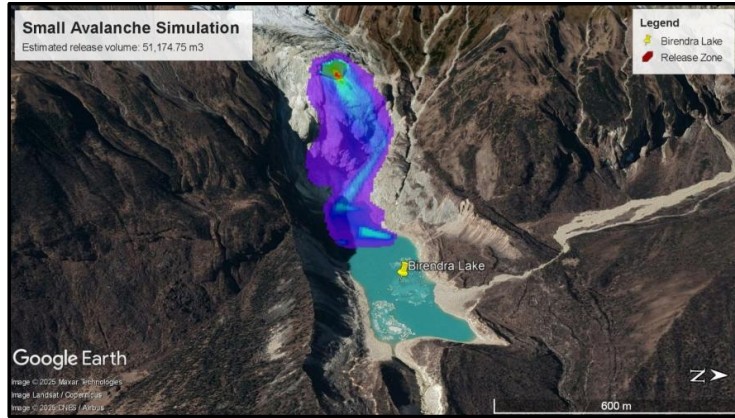


**Figure 6** Small avalanche simulation flow extent illustrating highly channelised flow following natural topographic corridors (© Google Maps 2025)


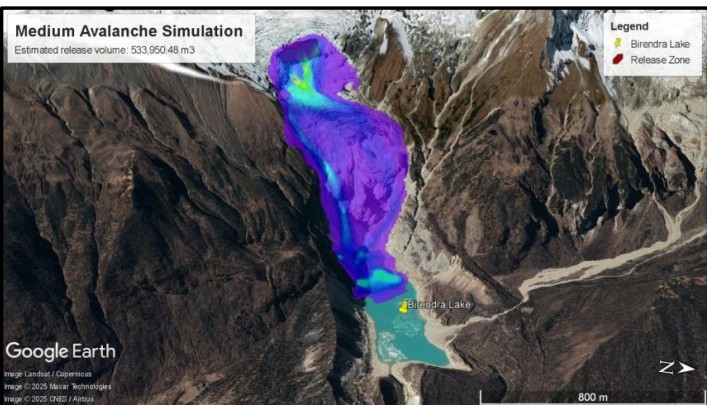


**Figure 7** Medium avalanche simulation flow extent demonstrating valley-confined flow with moderate lateral expansion (© Google Maps 2025)


Across all scenarios, the colour gradient (purple to yellow, indicating increasing flow height)
highlights flow concentration patterns and demonstrates how topographic controls influence
avalanche behaviour. Despite their different spatial patterns, all three scenarios demonstrate direct
impact potential on Birendra Lake, confirming that these source areas represent viable triggers for
cascading displacement floods and validating the susceptibility mapping approach used for release





zone identification.

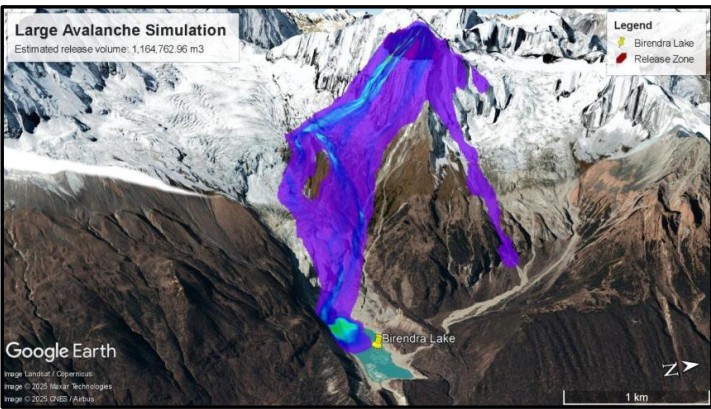


**Figure 8** Large avalanche simulation flow extent showing maximum lateral spreading and multi-
directional flow patterns (© Google Maps 2025)

**3.6 HEC-RAS Inflow Hydrographs for Various Avalanche Scenarios**

The outputs from the RAMMS simulations were translated into inflow hydrographs for the HEC-
RAS model, representing the initial flood pulse generated by avalanche impact and subsequent
water displacement. Following the methodology established by Chisolm and McKinney (2018) for
avalanche-induced lake displacement events, three displacement fractions (30%, 45%, and 60%)
of the deposited avalanche volume were applied to represent a range of plausible lake responses,
accounting for inherent uncertainties in impulse wave dynamics and lake-avalanche interaction
mechanisms. This approach acknowledges that displacement efficiency varies based on impact
velocity, avalanche density, and lake bathymetry, with the 45% displacement scenario serving as
the primary basis for downstream flood analysis. The resulting hydrographs for the small, medium,
and large scenarios are presented in Figures 9, 10 and 11.




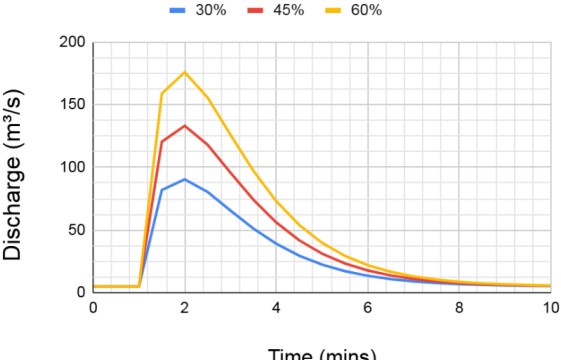


**Figure 9** Small avalanche scenario flood hydrograph showing characteristic impulsive discharge pattern with peak flows ranging from 90.4-175.9 m³/s across three displacement scenarios (blue = 30%, red = 45%, orange = 60% displacement fractions)


Small Scenario (Fig 9): For the smallest avalanche (51,200 m³ estimated release volume), the resulting flood hydrograph shows peak discharges ranging from 90.4 m³/s for a 30% displacement to 175.9 m³/s for a 60% displacement. The 45% displacement scenario generates a peak flow of 133.2 m³/s, representing a significant but localised flood event. While modest compared to larger scenarios, these flows exceed typical seasonal discharge variations and are sufficient to initiate downstream flooding with potential impacts on valley-floor infrastructure.

511

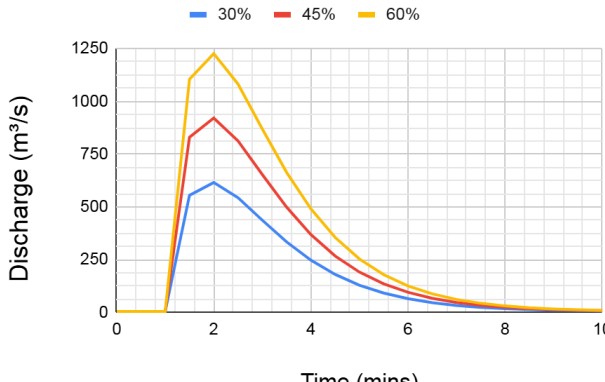

512

**Figure 10** Medium avalanche scenario flood hydrograph demonstrating substantial discharge amplification with peak flows ranging from 615.7-1,226.4 m³/s across displacement scenarios (blue = 30%, red = 45%, orange = 60% displacement fractions).

Medium Scenario (Fig 10): The medium avalanche scenario (534,000 m³ estimated release volume) produces a drastically larger flood pulse with peak discharges increasing substantially, ranging





from 615.7 m³/s (30% displacement) to a formidable 1,226.4 m³/s (60% displacement). The 45%
scenario peak of 921.0 m³/s marks a critical threshold where flood magnitude transitions from
localised impact to potentially catastrophic downstream effects. This scaling demonstrates the non-
linear relationship between avalanche volume and resultant flood severity, consistent with
observations from similar Himalayan GLOF events.

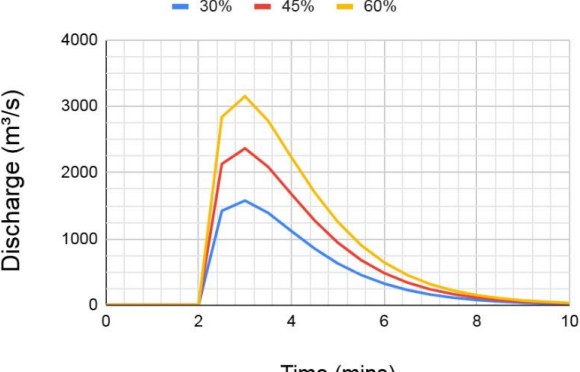


**Figure 11** Large avalanche scenario flood hydrograph showing catastrophic discharge potential
with peak flows ranging from 1,578.4 - 3,151.8 m³/s across displacement scenarios (blue = 30%,
red = 45%, orange = 60% displacement fractions)

Large Scenario (Fig 11): Representing a catastrophic failure from the upper glacier zones, the large
scenario (1,165,000 m³ estimated release volume) generates exceptionally high peak flows. The
discharge ranges from 1,578.4 m³/s (30% displacement) to a massive 3,151.8 m³/s (60%
displacement), with the 45% scenario peaking at 2,365.1 m³/s. Such discharge magnitudes are
comparable to major GLOF events documented in the Himalayan region and would be capable of
causing widespread and severe destruction to downstream communities and infrastructure.

**Temporal Characteristics and Warning Implications**
The hydrographs for all scenarios share a characteristic morphology: a rapid, single-peaked pulse
with a steep rising limb and a slightly less steep falling limb. This pattern is consistent with
impulsive, short-duration events where flood energy is released over minutes rather than hours,
aligning with documented avalanche-triggered GLOF behaviour from comparable Himalayan
systems (Sattar et al., 2022; Maharjan et al., 2024). The entire significant outflow for all scenarios
occurs within approximately 8-10 minutes, which is consistent with the 6-minute lake-emptying
time observed in the 2016 Gongbatongsha event (Sattar et al., 2022) and the 3-minute avalanche-





to-discharge timeframe documented at Imja Tsho (Lala et al., 2018). This temporal compression
underscores the limited time for downstream warning and emergency response. This rapid onset is
particularly critical for hazard management in the Manaslu region, as it eliminates traditional flood
warning lead times observed in conventional riverine flooding scenarios and necessitates pre-
positioned emergency response capabilities rather than reactive measures. The brief duration also
amplifies peak discharge rates, as the same displaced volume concentrated into shorter periods
generates significantly higher instantaneous flows, a phenomenon well-documented in avalanche-
triggered GLOF events across the Himalayas (Worni et al., 2014; Wang et al., 2018)

**Sensitivity Analysis and Hazard Assessment Implications**
The analysis of these inflow hydrographs demonstrates that the resultant flood magnitude is highly
sensitive to the initial avalanche volume and the assumed ice-water displacement efficiency. The
non-linear increase in peak discharge from Small (133.2 m³/s) to Large (2,365.1 m³/s) scenarios—
representing an 18-fold amplification despite only a 23-fold increase in avalanche volume—
highlights the critical importance of accurately identifying potential release volumes for hazard
assessment in glaciated mountain environments (Christen et al., 2010; Gabl et al., 2017). The
displacement fraction sensitivity is equally significant, with 60% displacement scenarios
generating 1.8-2.0 times higher peak flows than 30% scenarios across all avalanche magnitudes.
This sensitivity underscores the importance of continued research into avalanche-lake interaction
dynamics. It validates the conservative 45% displacement assumption adopted for primary hazard
modelling applications in this study, aligning with established Himalayan GLOF modelling
practices.

**3.7    Exposure analysis of Avalanche-induced flood scenarios at different sites**
The scenario-based exposure analysis of avalanche-induced Glacier Lake Outburst Floods
(GLOFs) at six different downstream sites offers critical insights into how the magnitude of such
events impacts the spatial extent of inundation, flood depth, and exposure to infrastructure. This
comparative study of Scenario 1 (small), Scenario 2 (medium), and Scenario 3 (large) highlights
significant differences in hazard levels across varied terrain settings and settlement patterns. By
assessing inundation patterns and interpreting flood behaviour at each site, this analysis contributes
to understanding both immediate and extended risks posed by GLOFs in vulnerable Himalayan
regions.





**Table 4** Lake Displacement Analysis

|  | Small | Med | Large |
|---|---|---|---|
| Total Avalanche Volume Outflow (m³) | 45,900 | 328,100 | 845,300 |
| Total Lake Volume Overspill (%) | 0.01 | 0.07 | 0.18 |

Table 4 demonstrates the direct relationship between avalanche magnitude and lake displacement volumes under the 45% displacement assumption. The Small scenario generates minimal outflow (45,900 m³, 0.01% of lake volume), while the Medium and Large scenarios produce substantially greater displacements of 328,100 m³ (0.07%) and 845,300 m³ (0.18%) respectively. Despite substantial volume increases, all scenarios displace less than 0.2% of total lake capacity, indicating overspill rather than complete drainage as the primary flood mechanism. This finding suggests that Birendra Lake would remain largely intact even under catastrophic avalanche impacts, with displaced water volumes serving as the primary driver of downstream flooding rather than complete lake breach scenarios.

The results of the HEC-RAS flood modelling are presented in Table 5. Moreover, as visualised in Figure 12, a clear progression of hazard severity corresponds to the magnitude of the initial avalanche. In Scenario 1 (Small), the flood is a low-impact but far-reaching event. It arrives at Site 1 (Samagaon) in 0.43 hours with a shallow depth of 0.96 m and takes 19.76 hours to reach the final site, Jagat. In contrast, the Scenario 3 (Large) flood wave is far more rapid and destructive, reaching Site 1 in only 0.15 hours and arriving at Jagat in just 4.6 hours. The increase in destructive potential is evident in the hydraulic data. At Site 1 (Samagaon), the maximum flood depth increases dramatically from 0.96 m in Scenario 1 to 12.69 m in Scenario 3, while the maximum velocity skyrockets from a manageable 1.94 m/s to a highly destructive 15.62 m/s. The inundation maps in Figure 12 visually confirms these findings. The most striking feature is the extensive, unconfined flooding at Site 1 (Samagaon) in Scenarios 2 and 3, where the inundation covers a wide portion of the valley floor where the settlement is located. While the flow becomes more channel-bound downstream, the maps clearly show significantly greater depths and widths for the larger scenarios, underscoring the severe risk posed to all downstream sites.



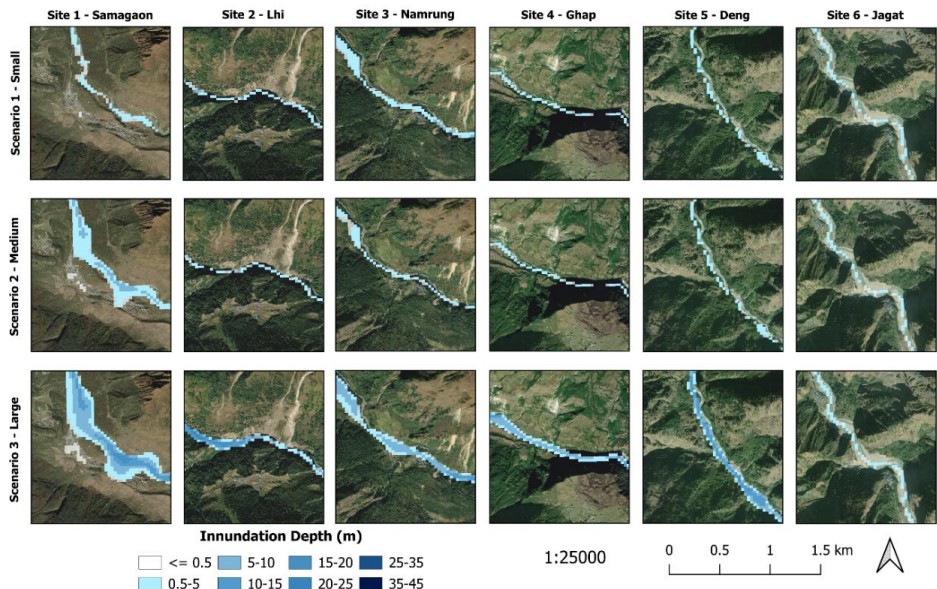

**Figure 12** Flood inundation depth maps for three avalanche-triggered scenarios across six downstream monitoring sites, shown using a color scale (© Google Maps 2025).

**Table 5** Hydraulic characteristics and flood arrival times for avalanche-triggered GLOF scenarios at downstream monitoring sites

| Sites | Flood arrival time (hrs) | | | Max flood depth (m) | | | Max flood velocity (m/s) | | |
|---|---|---|---|---|---|---|---|---|---|
| | Sc-1 | Sc-2 | Sc-3 | Sc-1 | Sc-2 | Sc-3 | Sc-1 | Sc-2 | Sc-3 |
| Site 1 | 0.43 | 0.15 | 0.15 | 0.96 | 5.82 | 12.69 | 1.94 | 9.49 | 15.62 |
| Site 2 | 5.04 | 1.02 | 0.5 | 2.47 | 2.47 | 16.09 | 1.52 | 1.83 | 8.24 |
| Site 3 | 6.76 | 1.82 | 0.7 | 0.56 | 2.3 | 11.92 | 1.86 | 2.01 | 6.04 |
| Site 4 | 9.41 | 3.32 | 0.96 | 0.73 | 0.83 | 11.9 | 1.14 | 1.38 | 2.26 |
| Site 5 | 13.01 | 6.19 | 1.45 | 0.57 | 0.57 | 11.13 | 2.49 | 2.55 | 4.97 |
| Site 6 | 19.76 | 12.84 | 4.6 | 0.86 | 1.04 | 1.22 | 0.93 | 0.93 | 1.37 |

Flow parameters derived from HEC-RAS 2D modeling showing flood arrival times (including initial avalanche travel time to lake), maximum flood depths, and peak velocities for three scenarios (Sc-1, Sc-2, Sc-3) across six sites from Samagaon (Site 1) to Jagat (Site 6).






### 3.8 Comparison with an Alternative Hazard Scenario: Dam Breach vs Lake Overspill

A critical comparison with the concurrent study by Poudel et al. (2025) provides essential context for understanding the spectrum of GLOF hazards at Birendra Lake. The fundamental distinction lies in triggering mechanisms and physical processes: Poudel et al.'s research models a hypothetical moraine dam breach scenario involving structural failure of the terminal moraine, whereas this study simulates avalanche-triggered lake overspill through displacement wave generation, which is a process directly analogous to the observed 21 April 2024 event. Peak discharge comparisons reveal the contrasting hazard magnitudes: dam breach scenarios generate extreme peak flows of 853-10,631 m³/s through rapid lake drainage, while overspill scenarios produce more moderate but still significant peaks of 133-2,365 m³/s through displacement-driven outflow. This order-of-magnitude difference reflects the distinct physical processes, where breach scenarios involve catastrophic structural failure, releasing large fractions of total lake volume, versus overspill events, displacing limited water volumes (less than 0.2% of lake capacity) while maintaining moraine integrity.

Temporal characteristics further distinguish these processes: dam breach events typically exhibit sustained high discharge over hours as the breach widens and lake drains, while avalanche-triggered overspill demonstrates impulsive, short-duration pulses consistent with the 8–10 minute significant outflow documented in this study. This temporal distinction has critical implications for warning systems, as overspill events provide even less reaction time than breach scenarios. Process probability assessments suggest that overspill events represent higher-frequency, moderate-impact hazards compared to the' lower-frequency, extreme-impact nature of complete dam failures. The April 2024 event demonstrates the immediate relevance of avalanche-triggered processes, while structural dam failure remains a longer-term concern requiring ongoing moraine stability monitoring. Both studies independently validate that Birendra Lake poses multi-modal GLOF threats requiring comprehensive risk management strategies. The overspill hazard documented here represents the immediate, observable threat that has already materialised, while Poudel et al.'s breach scenarios quantify potential future catastrophic risks under extreme conditions. This dual characterisation provides the hazard spectrum necessary for effective disaster risk reduction planning in the Manaslu region.





### 3.9 Uncertainty Discussion and Limitations

Uncertainty sources were identified using established guidelines for natural-hazard modelling. The main limitations are:

- Process simplifications – Avalanche-generated impulse waves are not simulated, and dam-erosion or sediment feedback are excluded, so peak discharge and arrival time may be misestimated.
- Terrain data (30 m DEM) – The grid smooths narrow channels and levees that steer flow, leading to location-specific depth and velocity errors.
- Lake geometry & roughness – Bathymetry is inferred from empirical area–volume curves, and Manning's $n$ values come from literature; both introduce unknown bias into the hydrograph.
- Scenario assumptions – A single, fixed displacement ratio and three deterministic avalanche sizes replace a full probabilistic ensemble, masking low-likelihood, high-impact events.
- Temporal factors – The modelling does not account for seasonal variations in lake levels or potential dam erosion processes that could modify flood characteristics.

## 4 CONCLUSION AND RECOMMENDATIONS

### 4.2 CONCLUSION

This research establishes the first comprehensive quantitative assessment of avalanche-triggered GLOF hazards at Birendra Lake, demonstrating that medium to large-scale ice avalanches ($\geq 5.3 \times 10^5$ m³) originating from the Manaslu Glacier pose a critical and imminent threat with the potential to trigger catastrophic overspill flooding. The integrated RAMMS-HEC-RAS modelling approach successfully simulated the complete hazard cascade from avalanche release to downstream flood propagation, validating the physically viable overspill mechanism observed during the April 21, 2024, event. All three simulated scenarios reached Birendra Lake with substantial mass retention ranging from 62% to 86%, while generating maximum flow velocities up to 72.8 m/s, demonstrating the high-energy nature of these cascading processes consistent with documented Himalayan avalanche behaviour.





The non-linear relationship between avalanche volume and flood severity is a critical finding for
hazard assessment applications. At the same time, the small-volume scenario (51,200 m³) produced
only minor downstream impacts; medium and large scenarios (534,000 m³ and 1,165,000 m³,
respectively) generated disproportionately severe flood waves that pose significant threats to
downstream communities. At Samagaon, the progression from small to large scenarios produces a
13-fold increase in maximum flood depths (0.96m to 12.69m) and an 8-fold surge in peak velocities
(1.94 m/s to 15.62 m/s), indicating critical threshold behaviour where moderate increases in
avalanche magnitude generate catastrophic downstream amplification.

The temporal compression dynamics represent the most critical finding for disaster risk reduction
applications. Flood waves reach Samagaon within 9-26 minutes, depending on avalanche
magnitude, with the large scenario generating arrival times comparable to the 6-minute lake-
emptying observed at Gongbatongsha and the 3-minute avalanche-to-discharge sequence
documented at Imja Tsho. This extreme temporal compression eliminates traditional flood warning
paradigms. It necessitates fundamental shifts toward pre-positioned emergency response rather
than reactive evacuation strategies, aligning with established patterns of impulsive, short-duration
GLOF behaviour across comparable Himalayan systems.

Despite methodological simplifications inherent in the displacement efficiency assumptions and
topographic resolution constraints, the modelling results demonstrate strong qualitative alignment
with observed characteristics from the April 2024 event, including rapid onset, overspill
mechanism, and downstream impact patterns. The conservative parameter selection (45%
displacement efficiency derived from established Himalayan GLOF modelling practices, literature-
based roughness values) suggests that results represent reasonable lower-bound estimates for
planning applications while maintaining physical consistency with documented cascading
processes.

These findings contribute essential quantitative evidence to the growing understanding of
cascading mountain hazards under accelerating climate change, with broader implications for
hundreds of similar glacier-lake systems throughout the Himalayas, where steep glacier termini
create comparable geometric conditions for avalanche-triggered processes. The study establishes
both immediate risk insights for the vulnerable Manaslu region and methodological contributions





to climate adaptation planning, providing a replicable framework for rapid hazard assessment in
data-scarce mountain environments and supporting the urgent need for comprehensive disaster risk
reduction strategies in one of the world's hazard-prone regions.
**4.3   RECOMMENDATIONS**
The recommendations of our study include:
a. Immediate implementation of a multi-component early warning system is essential,
incorporating glacier stability sensors, lake level monitoring, and automated downstream
alerts, given the extremely rapid flood arrival times (minutes to critical settlements).
b. Hazard-based land use planning should implement the detailed flood inundation maps as
regulatory tools, prohibiting new construction within high-hazard zones and assessing
existing infrastructure vulnerability, particularly in Samagaun's vulnerable valley floor area.
c. Community preparedness programs must be developed with targeted risk communication and
evacuation procedures integrated into existing disaster risk reduction frameworks.
d. Future research priorities include conducting detailed bathymetric surveys of Birendra Lake
to reduce critical uncertainties, acquiring high-resolution topographic data, developing
explicit impulse wave generation models, and integrating geotechnical dam stability analysis.
e. Long-term monitoring frameworks combining remote sensing with in-situ measurements
should be established to track system evolution and provide validation data for model
refinement.
f. Extending the integrated modelling approach to other high-risk glacial lake systems
throughout the Himalayas to support regional climate change adaptation planning and
develop standardised protocols for rapid hazard assessment in data-scarce environments.
**CODE AVAILABILITY**
Not applicable
**DATA AVAILABILITY**
Contact the corresponding author for more information on access to datasets.



**AUTHOR CONTRIBUTIONS**

Mohan, Ragini, Rijan and Sujan designed the study. Ragini and Sujan contributed to data collection. Ragini and Sujan performed the data analysis and interpretation. Mohan, Ragini, Rijan, and Sujan prepared the manuscript with contributions from all co-authors. Mohan and Rijan supervised the research.

**COMPETING INTERESTS**

The contact author has declared that none of the authors has any competing interests.

**ACKNOWLEDGEMENTS**

We thank the National Trust for Nature Conservation (NTNC)-Manaslu Conservation Area Project (MCAP) for their generous financial support. The grant provided by NTNC-MCAP played a crucial role in completing this project. In addition, we would like to express our sincere gratitude to the RAMMS development team at the WSL Institute for Snow and Avalanche Research SLF for kindly providing the RAMMS software license.

**FINANCIAL SUPPORT**

This research has been supported by the National Trust for Nature Conservation- Manaslu Conservation Area Project (NTNC-MCAP).

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

**APPENDICES**
**Appendix A**
Small simulation showing maximum height, velocity, and pressure

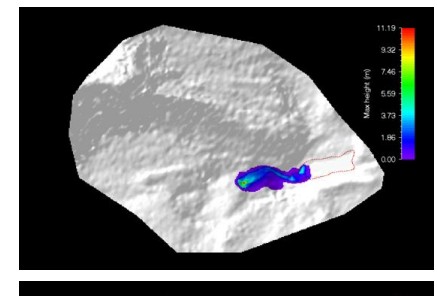
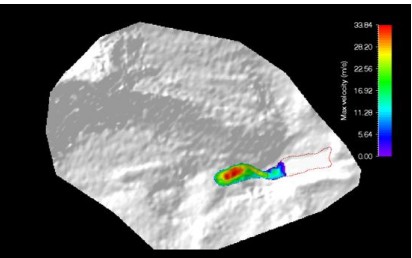

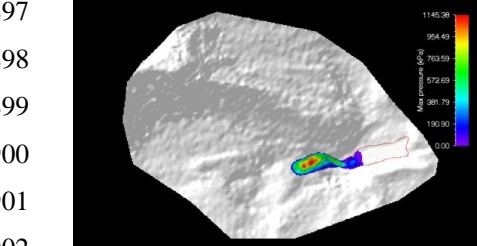







**Appendix B**

Medium simulation showing maximum height, velocity, and pressure

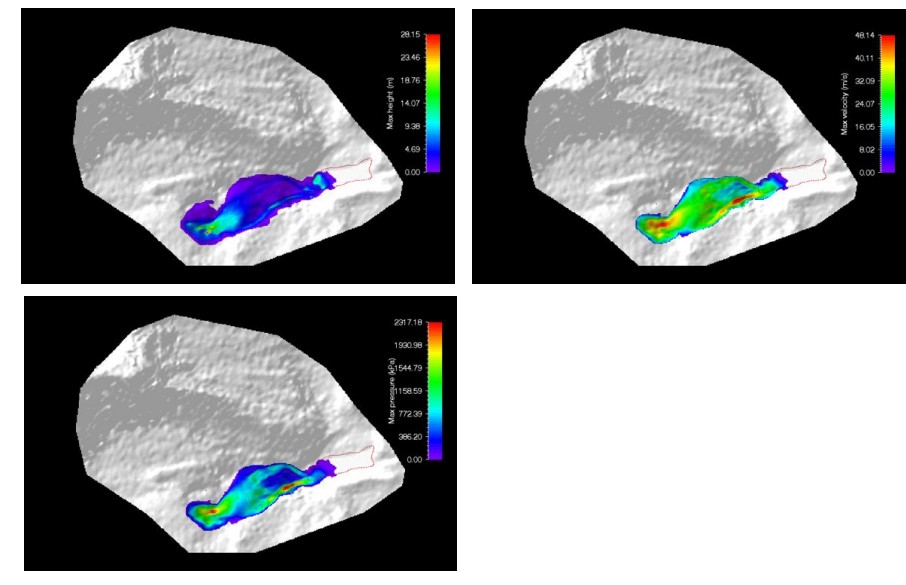

**Appendix C**

Large Simulation showing maximum height, velocity, and pressure

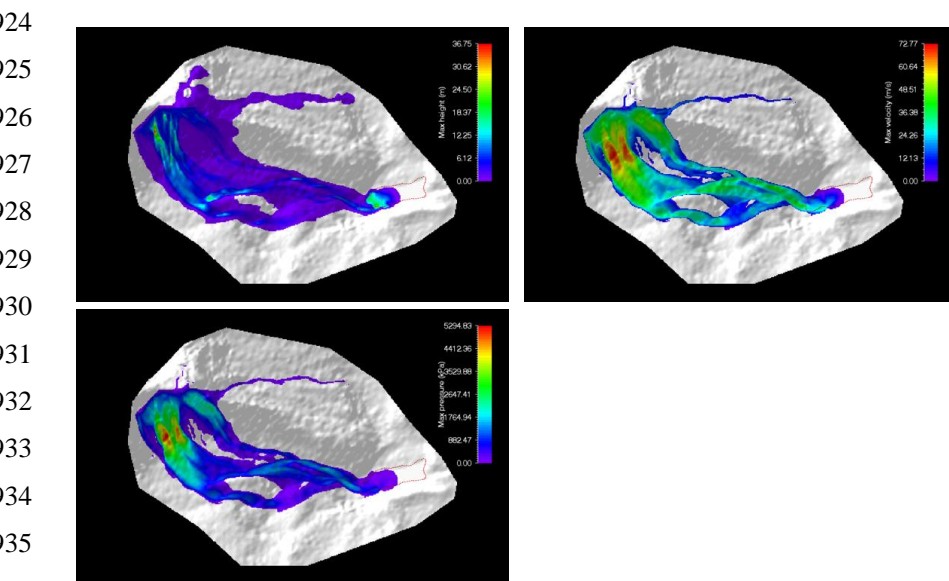



937 **Appendix D**

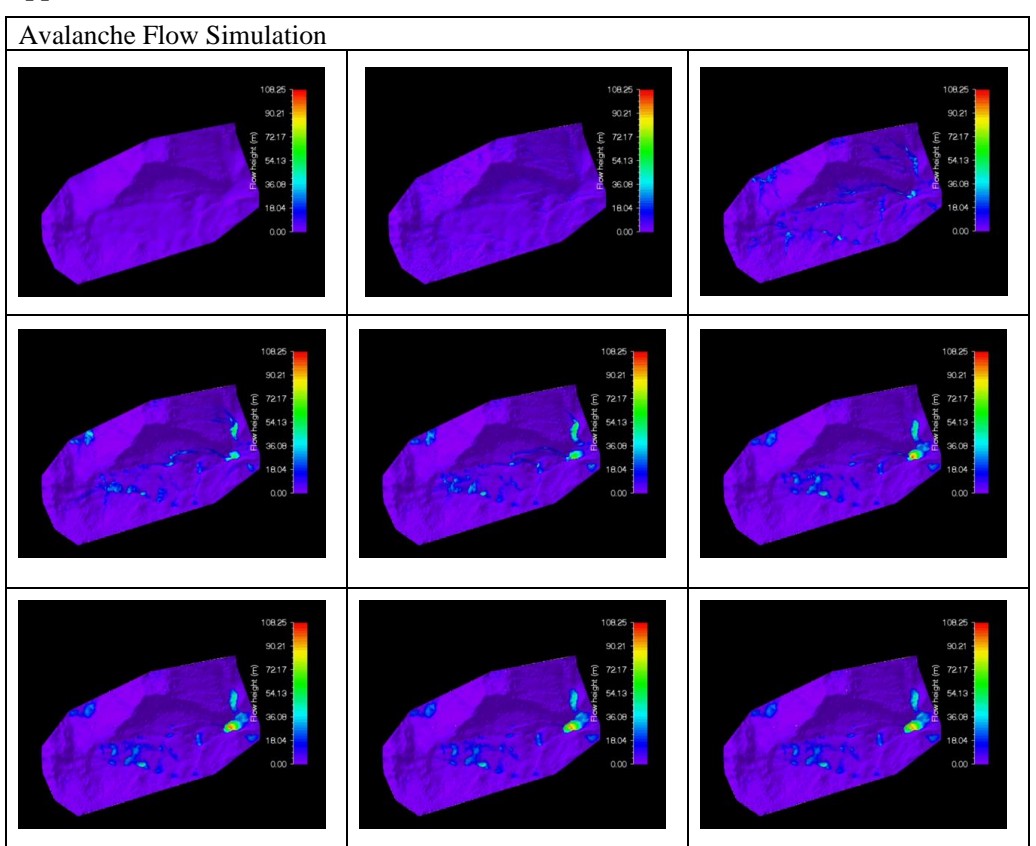

938