# Peer review of "Simulating avalanche-triggered lake overspill and downstream impacts at"

_EGUsphere, 2025_

## Author Comment (AC1)

**Response to Reviewer Comments**

**Manuscript ID:** egusphere-2025-4454
**Title:** Simulating avalanche-triggered lake overspill and downstream impacts at Birendra Lake using RAMMS and HEC-RAS
**Authors:** Sujan Thapa, Ragini Vaidya, Mohan Bahadur Chand, Rijan Bhakta Kayastha

Dear Editor and Reviewers,

We thank the reviewers and community members for their insightful and constructive feedback on our manuscript. We have addressed the comments, particularly regarding the manuscript structure, technical terminology, and overspill volume calculations.

Our point-by-point responses are provided in the table below:

**Response to Referee #2 (RC2)**

| ID | Reviewer Comment | Author Response |
|---|---|---|
| RC2.1 | *Structure – the study suffers from rather unusual structure (intro section seems unusually long; separate discussion section is definitely needed; results section should be structured according to objectives / methods used and needs to only present results; study area section is not a part of the methods, ...); as a result, it is difficult to read it and understand the work done* | Manuscript restructured for clarity:

● Introduction: Condensed to focus on core research gaps.
● Study Area: Moved to Section 2, preceding Materials and Methods.
● Results and Discussion: Separated into distinct sections.
● Alignment: Results sub-sections now follow the sequence of the methodology and study objectives. |
| RC2.2 | *Methods – the use of some of the methods doesn't seem suitable / justified (the Bühler et al. 2013 methodology was developed for snow avalanches but here the authors model ice-avalanches (different mechanisms / processes); since the time of famous Huggel et al., 2004 lake area-volume scaling relationship, many other Himalaya-focusing methods for estimating lake volume have been developed and published since, providing better performance).* | Acknowledged. We replaced the Huggel et al. (2004) relationship with the Himalayan-specific empirical equation from Zhang et al. (2023). This yields a more regionally accurate and slightly higher volume estimate.

Clarified that the Bühler et al. (2013) method was used strictly as a preliminary GIS-based filtering tool to identify potential release zones based on terrain characteristics. |
| RC2.3 | *Modelling parameters and assumptions – parameters that are used need justification other than the use in previous studies (e.g. Manning) - please not only use the values but also discuss the performance / sensitivity evaluations from previous studies;* | A formal discussion on sensitivity based on Poudel et al. (2025) which conducted the Manning's n sensitivity analysis is provided. |

| ID | Reviewer Comment | Author Response |
|---|---|---|
| | *what is the unit of curvature 50? and standard deviation of terrain roughness 15m? - please check;* | We have clarified the modeling parameters and units. The curvature value of 50 is unitless (dimensionless coefficient) following the Bühler et al. (2013) algorithm. The terrain roughness value of 15m represents the standard deviation of elevation (in meters). |
| | *Calculated % of released volume are not correct (800,000 m3 of a large scenario is not 0.18% of the lake volume);* | Correction: The overspill range is 1.0% to 18% of total lake volume, not 0.01% - 0.18%. This decimal error will be corrected throughout the manuscript. |
| | *Why is x axis and the shape of all hydrographs the same, regardless the scenario? How they were created (considering dam overtopping mechainsm and lake dynamics, the shape would be a series of waves rather than one several minutes-lasting wave)?* | We used a single-peaked hydrograph to capture the primary overspill pulse and the peak discharge. This was generated using a discharge multiplier to simulate a rapid, impulsive release. We focused on the initial wave because it defines the maximum flood depth and earliest arrival times at downstream sites. While real events have multiple waves, modeling the secondary wave series was beyond the current scope of this hazard assessment. |
| RC2.4 | *Terminology – the terminology is not used properly (Exposure – the authors write about exposure in abstract and text but no exposed elements are mapped at the end, flow depth is not a characterics of exposure; vulnerability – the lake is not vulnerable to avalanches but prone to avalanche impacts, ...) undermining the work done* | "Exposure" was replaced with "hazard assessment" throughout. However, We have tried to show the impacts in terms of population, settlements and infrastructures and discuss accordingly.

"Vulnerability" corrected to "proneness" or "susceptibility" to reflect physical conditions accurately. |
| RC2.5 | *The 2024 GLOF event mentioned in the intro provides an opportunity for evaluation the performance of used models; however, no detailed info of this event and its impacts nor further analysis are provided* | Added a comparison between simulation results and documented findings from recent literature regarding the 2024 event to evaluate model performance. |
| RC2.6 | *Recommendations - the list of recommendations are predominantly general and tru for all potential GLOF sites; what site-specific recommendations can be derived from the results of the study?* | Specific recommendations will be incorporated based on revised discussion on the manuscripts. |

---

## Author Comment (AC2)

**Response to Reviewer Comments**

**Manuscript ID:** egusphere-2025-4454
**Title:** Simulating avalanche-triggered lake overspill and downstream impacts at Birendra Lake using RAMMS and HEC-RAS
**Authors:** Sujan Thapa, Ragini Vaidya, Mohan Bahadur Chand, Rijan Bhakta Kayastha

Dear Editor and Reviewers,

We thank the reviewers and community members for their insightful and constructive feedback on our manuscript. We have addressed the comments, particularly regarding the manuscript structure, technical terminology, and overspill volume calculations.

Our point-by-point responses are provided in the table below:

**Response to Referee #1 (RC1)**

| ID | Reviewer Comment | Author Response |
|---|---|---|
| RC1.1 | *The manuscript requires significant restructuring to clearly convey the objective, methodology, results, limitations, and conclusions of the study. Currently, many technical terms, topics, and section headings are introduced abruptly without prior explanation or context, making it difficult for the reader to follow the narrative.* | Accepted. The manuscript has been organized to ensure a logical flow and alignment between the objectives, methodology, and results. We have clarified the technical terms and ensured that the headings are introduced with proper context. |
| RC1.2 | *There is a noticeable lack of alignment between the study's stated objectives, the flowchart presented in Figure 3, the methodology described in Section 2, and the Results and Discussion section. These components should be revised to ensure coherence and logical flow throughout the manuscript.* | Accepted. The stated objectives and methodology have been revised to align with the study flow chart (Figure 3), ensuring consistency throughout the sections. |
| RC1.3 | *It is suggested to restructure the methodology into two separate sections as follows:*

● *Section 2 – Study Area*

● *Section 3 – Materials and Methods, with the following subsections:* | Accepted. We have restructured the methodology as suggested. |

| ID | Reviewer Comment | Author Response |
|---|---|---|
| |     ○ *1 Avalanche Susceptibility Mapping*

     ○ *2 Glacial Lake Volume Estimation*

     ○ *3 RAMMS Modeling*

     ○ *4 HEC-RAS Modeling*

     ○ *5 Exposure Analysis*

 *Each subsection should clearly describe its role in the study, including the input parameters used and references where applicable. A similar structure should also be followed in the Results and Discussion section for consistency. Subsections can be further divided as needed to highlight important components.* | |
| RC1.4 | ***Results and Discussion Section***

 *The Results and Discussion section should focus solely on the outputs and findings of the present study. Currently, this section often reads like an extended version of the methodology. The descriptive elements already covered in the methodology should not be repeated. Instead, this section should include clear interpretation and analysis of the model outputs, supported by relevant figures and tables.* | Accepted. This section has been revised to focus on the interpretation and analysis of the model outputs. |
| RC1.5 | *There are two unnumbered subsections presented under Results and Discussion which were not introduced or discussed in the methodology:*

     ● ***Temporal Characteristics and Warning Implications***

     ● ***Sensitivity Analysis and Hazard Assessment Implications*** | Accepted. These subsections have been formally integrated into the manuscript structure and discussed in the methodology. |

| ID | Reviewer Comment | Author Response |
|---|---|---|
| RC1.6 | *These need to be formally integrated into the manuscript structure. Additionally:*

 • *The source of the outflow timing estimate (8–10 minutes) should be clarified. Including a figure showing the progression of the flood from initiation to downstream points would greatly enhance clarity.*

 • *The parameters considered for sensitivity analysis should be clearly identified and their impact discussed.* | Accepted. The 8–10-minute estimate represents the duration of the primary overspill pulse at the lake outlet, as derived from our HEC-RAS inflow hydrographs. We have expanded the discussion of flood progression in Section 4.5. Rather than adding a new figure, we have referenced the arrival times provided in Table 5, which detail the temporal progression of the flood from initiation to the six downstream monitoring sites.

 A formal discussion on sensitivity based on Poudel et al. (2025) which conducted the Manning's n sensitivity analysis is provided. |
| RC1.7 | *Section 3.7, titled Exposure Analysis of Avalanche-Induced Flood Scenarios at Different Sites, currently discusses flood arrival time, maximum flow depth, and velocity. However, it does not include any actual **exposure analysis**, which by definition involves identifying and quantifying the **elements at risk** (e.g., population, infrastructure, land use). This section should be revised to include or refer to such analysis, or the section title should be changed to accurately reflect its content.* | The discussion is revised with incorporation of the impacts based on secondary data (population, infrastructure, settlements). |
| RC1.8 | ***Figures***
 *The following revisions are suggested for the figures:*

 • *The **Study Area Map** should be labeled as **Figure 1**.*

 • ***Figures 4 and 5** present similar information except for the three different release zones; these could be combined into a single comparative figure for better clarity and reduced redundancy.*

 • ***Figures 6, 7, and 8** ( RAMMS simulation outputs) can be consolidated into one* | Figure labeling and organization have been updated: the Study Area Map is now Figure 1; Figures 4 and 5 are merged to show susceptibility and release zones together; and RAMMS simulation outputs (originally Figs 6,7, & 8) are consolidated into a single multi-panel figure with panel labels, parameter-specific color bars, and descriptive captions. |

| ID | Reviewer Comment | Author Response |
|---|---|---|
| | *composite figure with clearly labeled panels. Each panel should include color bars indicating the dynamic parameter being shown (e.g., height, velocity, momentum). Be sure to mention that these represent RAMMS outputs in the figure caption.* | |
| RC1.9 | **RAMMS:** *What is the basis and background for identification, demarcation and consideration of the release zones and the release depth?* | Release zones were identified using the susceptibility map; the medium scenario source area specifically aligns with zones identified by Maharjan et al. (2024). A consistent 5.0 m release depth was applied based on ice failure depths used in Sattar et al. (2021) and Mandal et al., (2025) for similar Himalayan contexts. |
| RC1.10 | *If the main implementation of the RAMMS model has been done to estimate the volume of material reaching at the lake, then it is not clear why the 2nd/3rd scenario was proposed/assumed. Different scenario might have also formulated using different release depth and initial volume at the scenario-I (Small).* | Three distinct release zones were selected based on the susceptibility map to evaluate the influence of varying flow-path topography on lake impact. The results confirm that release location is as critical as volume; for instance, the 'Small' scenario followed a more direct, channelized path, achieving higher deposition efficiency than the 'Medium' scenario.

Section 3.3 has been updated to clarify that these scenarios test topographical path variability rather than volume scaling at a single site. |
| RC1.11 | **Input Data:** *Make a table to show all the input data used in this work. (DEM resolution, time, frictional parameters, entrainment (if any), release condition (block/hydrograph) for RAMMS & HECRAC.* | Accepted. A comprehensive Input Data Summary Table has been added as Appendix E (Table E1). This table consolidates all parameters used in the coupled modeling chain, including DEM resolutions (ALOS PALSAR 12.5m and SRTM 30m), RAMMS frictional parameters ($\mu=0.12, \xi=1000m/s$), Manning's $n(0.06)$, and the specific release conditions for RAMMS and HEC RAS were added. |
| RC1.12 | **HEC RAS:** *Why was a Manning's n value of 0.06 considered appropriate for the modelled Himalayan stream reach?* | A spatially uniform Manning's n value of 0.06 was adopted for the entire 2D flow area. This value was selected as a conservative composite roughness to reflect the high energy dissipation across both the boulder-strewn channel and the rugged, debris-covered valley floor. While studies like Poudel et al. (2025) utilized variable roughness values, our use of a single, |

| ID | Reviewer Comment | Author Response |
|---|---|---|
| | | higher value ensures a "safe-side" hazard assessment. |
| RC1.13 | *Why was roughness calibration considered limited or not performed in detail for this model?* | This is assumed as limitation for the data scarce region like High Mountain Himalayas and could be future scope. |
| RC1.14 | *How can sensitivity analysis be incorporated within HEC-RAS modeling to improve flood hazard assessment?* | Various parameters used in HECRAS, such as roughness coefficient, river geometry, glacial lakes and dam characteristics etc can be incorporated for sensitivity analysis, if all the data values are known. |

| CC1.1 | *For figures 9, 10 and 11, merge them and describe giving sub-number* | Figures 9, 10, and 11: Merged into a single composite figure with sub-number. |
|---|---|---|
| CC1.2 | *L715 cite the published paper not the preprint*
*Banerjee A, Meadows EM, Yadav N, et al. Glacier and glacial lake dynamics from 1990 to 2024 and their impact on flood hazard in the central Nepal Himalaya[J]. Journal of Mountain Science: 2025, 22:1926-1943. 10.1007/s11629-024-9298-0*
*Chaulagain M, Chand MB, Pradhananga D, et al. Recurring avalanche hazards at Birendra Lake, Manaslu region: Interdisciplinary insights from the April 21, 2024, avalanche event[J]. 2025, 7:59-77.*
*Khadka N, Zheng G, Chen X, et al. An ice-snow avalanche triggered small glacial lake outburst flood in Birendra Lake, Nepal Himalaya[J]. Natural Hazards: 2024, 121:6357-6365.*
*https://doi.org/10.1007/s11069-024-07014-0*
*Poudel U, Gouli MR, Hu K, et al. Multi-breach GLOF hazard and exposure analysis of Birendra Lake in the Manaslu Region of Nepal[J]. Natural Hazards Research: 2025. 10.1016/j.nhres.2025.03.007*

*Citation:*
*https://doi.org/10.5194/egusphere-2025-4454-CC1* | Updated all citations from preprints to the final published versions for Banerjee et al. (2025), Chaulagain et al. (2025), Khadka et al. (2024), and Poudel et al. (2025). |

---

## Author Comment (AC3)

**Response to Reviewer Comments**

**Manuscript ID:** egusphere-2025-4454
**Title:** Simulating avalanche-triggered lake overspill and downstream impacts at Birendra Lake using RAMMS and HEC-RAS
**Authors:** Sujan Thapa, Ragini Vaidya, Mohan Bahadur Chand, Rijan Bhakta Kayastha

Dear Editor and Reviewers,

We thank the reviewers and community members for their insightful and constructive feedback on our manuscript. We have addressed the comments, particularly regarding the manuscript structure, technical terminology, and overspill volume calculations.

Our point-by-point responses are provided in the table below:

**Response to Community #1 (CC1)**

| ID | Reviewer Comment | Author Response |
|---|---|---|
| CC1.1 | *I am happy to read the manuscript by Thapa et al. which have used RAMMAS modeling. Initially, I anticipated this research to be a reconstruction of the 2024 Birendra Lake GLOF event, however, it appears to be focused on a hazard assessment instead. This distinction should be made explicit to avoid potential misunderstandings.* | Thanks for your time and constructive feedbacks. We agree with this distinction. The manuscript will be revised to explicitly frame the study as a hazard assessment rather than a reconstruction of the April 2024 event. Clarifications have been added to the Abstract and Introduction to ensure this focus is clear to the reader. |
| CC1.2 | *Given the numerous published studies after the Birendra GLOF event (Khadka et al., 2024; Banerjee et al., 2025; Chaulagain et al., 2025; Poudel et al., 2025), it is essential for this study to acknowledge these prior works to build up the study and articulate what new insights or findings it brings to show the credibility.* | Yes, you have rightly mentioned and all the previous studies are acknowledged. Sections have been revised with sufficient discussion to acknowledge these studies and highlight our focus on the hazard process chain. |
| CC1.3 | *My main concern is that the authors claim 0.01-0.18% of total lake volume 4.7×106 m3 (=8460 m3 max) is spilled from lake which seems speculative and small when comparing to downstream discharges and time given in the manuscript. Please make me clear if I missed anything. Further, impact/exposure assessment as shown in Figure 3 has not been conducted.* | Correction: The overspill range is 1.0% to 18% of total lake volume, not 0.01% - 0.18%. This decimal error is corrected throughout the manuscript. |
| CC1.4 | *L49 What does ice-debris avalanche mean? The feeding glacier is a clean glacier* | Correction: Term changed to "ice avalanche". |

| ID | Reviewer Comment | Author Response |
|---|---|---|
| CC1.5 | *L50 vulnerability to susceptibility* | Correction: Term changed to "susceptibility". |
| CC1.6 | *L52 I think this event is a small event and has not affected or had impacts on downstream community/settlements* | Clarified that while the event was small-scale, it destroyed the bridge connecting Samagaun and Samdo. |
| CC1.7 | *L55-L58 cite previous studies that have investigated this event* | References added, including Maharjan et al. (2024), Khadka et al. (2024), Chaulagain et al. (2025), and Banerjee et al. (2025). |
| CC1.8 | *L61 two semi colon* | Corrected |
| CC1.9 | *L94 check the exact cause of the 2016 Gongbatongsaco GLOF* | Corrected. The 2016 Gongbatongsha GLOF was triggered by heavy precipitation causing a slope failure and debris deposition into the lake, leading to a moraine dam breach and rapid drainage. |
| CC1.10 | *L119 mention the glacial lake* | Corrected. Lake Palcacocha specifically mentioned. |
| CC1.11 | *L134 once the full form is abbreviated it is wise to use abbreviation throughout the ms* | Corrected. Abbreviations now used consistently following first mention. |
| CC1.12 | *L145 use another term for vulnerability* | "Vulnerability" changed to "proneness." |
| CC1.13 | *L167 glacier detach: is it recently or since the end of LIA? Prove if its recent from satellite imagery* | "Recently" removed to avoid temporal ambiguity regarding the glacier detachment. |
| CC1.14 | *Study area is poorly described. It would be great to know the climatology, glaciers, glacial lakes and their characteristics in the region. Why studying such event is important in the region?Authors could have used also reviewed more published studies in Nepal Himalaya for structuring the background/introduction and for study area (some maybe https://doi.org/10.1007/s10113-023-02142-y, https://doi.org/10.3390/rs9070654, https://doi.org/10.3390/rs10121913)* | The Study Area has been expanded to include regional climatology and glacier characteristics. The suggested published studies were referenced. |
| CC1.15 | *L226 which high resolution DEM?* | Clarified the use of ALOS 12.5m DEM. |
| CC1.16 | *L242 in repeated* | Redundant text removed. |
| CC1.17 | *L285 et al. is missing in the reference when there are more than 2 authors* | Citations corrected to include "et al." |
| CC1.18 | *Table 1 How was depth determined? Is it mean depth or max depth?* | Table 1: Depth removed; it represented mean depth and was non-essential for the final analysis. |
| CC1.19 | *L346 Why not to use HMA DEM of better resolution than ALOS?* | L346: ALOS DEM was used instead of HMA DEM because it contained fewer data voids in the glacierized sections, ensuring better simulation stability in RAMMS. |

| ID | Reviewer Comment | Author Response |
|---|---|---|
| CC1.20 | *L348 mention the weblink or cite it* | Citation and weblink for Adventure Consultants added. |
| CC1.21 | *For figures 9, 10 and 11, merge them and describe giving sub-number* | Figures 9, 10, and 11: Merged into a single composite figure with sub-number. |
| CC1.22 | *L715 cite the published paper not the preprint*
*Banerjee A, Meadows EM, Yadav N, et al. Glacier and glacial lake dynamics from 1990 to 2024 and their impact on flood hazard in the central Nepal Himalaya[J]. Journal of Mountain Science: 2025, 22:1926-1943. 10.1007/s11629-024-9298-0*
*Chaulagain M, Chand MB, Pradhananga D, et al. Recurring avalanche hazards at Birendra Lake, Manaslu region: Interdisciplinary insights from the April 21, 2024, avalanche event[J]. 2025, 7:59-77.*
*Khadka N, Zheng G, Chen X, et al. An ice-snow avalanche triggered small glacial lake outburst flood in Birendra Lake, Nepal Himalaya[J]. Natural Hazards: 2024, 121:6357-6365. https://doi.org/10.1007/s11069-024-07014-0*
*Poudel U, Gouli MR, Hu K, et al. Multi-breach GLOF hazard and exposure analysis of Birendra Lake in the Manaslu Region of Nepal[J]. Natural Hazards Research: 2025. 10.1016/j.nhres.2025.03.007*

*Citation: https://doi.org/10.5194/egusphere-2025-4454-CC1* | Updated all citations from preprints to the final published versions for Banerjee et al. (2025), Chaulagain et al. (2025), Khadka et al. (2024), and Poudel et al. (2025). |